# Recent Advances in Camel Milk Processing

**DOI:** 10.3390/ani11041045

**Published:** 2021-04-08

**Authors:** Gaukhar Konuspayeva, Bernard Faye

**Affiliations:** 1UMR SELMET, CIRAD-ES, 34398 Montpellier, France; konuspayevags@hotmail.fr; 2Department of Biotechnology, Faculty of Biology and Biotechnology, Al-Farabi Kazakh National University, Almaty 050040, Kazakhstan

**Keywords:** camel, milk technology, pasteurization, cheesemaking, powder milk, fermentation

## Abstract

**Simple Summary:**

The camel milk market was limited for a long time by its almost exclusive self-consumption use in nomadic camps. Significant development has been observed for the past two or three decades, including internationally, boosted by its reputation regarding its health effects for regular consumers. Such emergence has led the stakeholders in the sector to offer diversified products corresponding to the tastes of increasingly urbanized consumers, more sensitive to “modern” products. Thus, traditionally drunk in raw or naturally fermented form, camel milk has undergone unprecedented transformations such as pasteurization, directed fermentation, cheese or yoghurt processing, and manufacture of milk powder for the export market. However, the specific characteristics of this milk (composition, physical properties) mean that the technologies applied (copied from technologies used for cow milk) must be adapted. In this review, some technological innovations are presented, enabling stakeholders of the camel milk sector to satisfy the demand of manufacturers and consumers.

**Abstract:**

Camel milk is a newcomer to domestic markets and especially to the international milk market. This recent emergence has been accompanied by a diversification of processed products, based on the technologies developed for milk from other dairy species. However, technical innovations had to be adapted to a product with specific behavior and composition. The transformation of camel milk into pasteurized milk, fermented milk, cheese, powder, or other products was supported, under the pressure of commercial development, by technological innovations made possible by a basic and applied research set. Some of these innovations regarding one of the less studied milk sources are presented here, as well as their limitations. Technical investigations for an optimal pasteurization, development of controlled fermentation at industrial scale, control of cheese technology suitable for standardized production, and improvements in processes for the supply of a high-quality milk powder are among the challenges of research regarding camel milk.

## 1. Introduction

For a long time, only fresh camel milk was consumed by pastoralists and was regarded as a gift for the hosts coming under the tent of the nomads. Consequently, it was not considered a commodity and its sale was often taboo. Moreover, it did not undergo any transformation, except for fermentation [1] to prolong its shelf life in desert conditions where the cold chain could not be respected. The introduction of camel milk to the regular market at a national or even international level is a recent feature [2]. Such development of the camel milk market was concomitant with a deepening of the knowledge of its fine composition [3] and of its transformation processes, allowing the marketing of a more diversified dairy product [4]. Recent findings are effectively available, allowing this important product renowned for its true or supposed “medicinal” virtues [5] to remove camel milk from marginality. Indeed, the production of camel milk at a world level is experiencing a considerable annual growth, exceeding 8% in the period 2009–2019 [6], testifying to the growing interest for this product.

Moreover, despite its proximity with cow milk in term of gross composition (fat, protein, lactose, and mineral proportions), camel milk shows many specificities [3] such as functional proteins, predominant medium-chain fatty acids in fat matter, low lactose intolerance, and richness in vitamin C or iron, among others. These proper characteristics have an obvious impact on the “behavior” of camel milk during processing.

Thus, the present paper proposes the state of the art regarding knowledge on camel milk processing by focusing on four main dairy products having different success on the market, i.e., pasteurized milk, fermented milk, camel cheese, and camel milk powder. Other products will rapidly emerge. In addition to the review regarding the current status of knowledge on camel milk processing, the more recent studies (2019–2020) focusing on this topic are presented in Table 1.

## 2. Pasteurized Milk

### 2.1. Current Global Conditions for Pasteurization of Camel Milk

Pasteurization of camel milk is a commonly used technique in camel countries. Nevertheless, the conditions of pasteurization implemented by each holder were often decided without considering the specificity of camel milk, with the rules being mainly based on the standard issued for pasteurization of cow milk. Some data regarding the conditions of camel milk pasteurization in scientific literature are quite variable: 60 °C for 30 min [33]; 75 °C for 15 s [34]; 63 °C for 30 min [35,36]. At the same time, many private companies in the United Arab Emirates (UAE), Saudi Arabia, Mauritania, Kazakhstan, Algeria, Tunisia, Morocco, and Niger are producing and selling pasteurized camel milk on the local market. All of these companies apply different conditions for pasteurization. It is important to mention that national/regional/international standards are not yet elaborated for camel milk or they are simply copied from cow milk. In some camel countries, no standard is proposed by the government or, at least, it is suggested to apply the same conditions as for bovine milk.

### 2.2. Indicators of Camel Milk Pasteurization

The pasteurization procedure for camel milk should have its own conditions and indicators. Indeed, a preliminary study showed that alkaline phosphatase (ALP), traditionally used for cow milk [37], was not a convenient indicator of successful pasteurization of camel milk, because camel ALP is heat-resistant, still showing activity at 90 °C [38]. Loiseau et al. [39] suggested using glutamyltranspeptidase (75 °C for 30 s) or leucine arylamidase (75 °C for 28 s or 80 °C for 7 s) as an indicator of pasteurization for camel milk. If camel milk must be pasteurized at 72 °C for 20 min, the most appropriate indicator could be gamma-glutamyl transferase (GGT) according to Wernery et al. [40]. However, later in 2011, Lorenzen et al. [41] concluded that GGT was still present in pasteurized camel milk, whereas lactoperoxidase (LPO) could be a more appropriate indicator of pasteurization. Tayefi-Nasrabadi et al. [42] confirmed that camel LPO was less heat-resistant than bovine LPO. Until now, no sufficiently in-depth studies have been done in this field, although pasteurized camel milk was introduced to the international market. Such doubts on a convenient indicator of pasteurization for camel milk are a constraint for the establishment of an international standard. Thus, the pasteurization of camel milk at an industrial scale is possibly achieved in the wrong way and its heat treatment may be incorrect.

### 2.3. Impact of Pasteurization on Physical Properties of Camel Milk

The characteristics of camel milk flow in pipes during the pasteurization process, the cleaning procedures, and the conditions of transfer and pumping also need to be studied. The behavior of fat globules and casein micelles in camel milk is different from that in cow milk, presenting an absolute viscosity of 1.72 mPa·s at 20 °C vs. 2.04 mPa·s at the same temperature in bovine milk. Thus, in dairy plants, the camel milk should not necessarily be transferred at 20 °C. According to Kherouatou et al. [43], camel milk in fresh conditions (without acidification) can resist some mechanical strains without changing its microstructure. The apparent viscosity was quite stable (between 1.6 and 2.0 mPa·s) at pH values between 5.2 and 6.7. Therefore, camel milk can be manipulated in dairy plants, e.g., pumping, agitation, skimming, homogenization, and bottling, for producing a pasteurized product.

Few studies have focused on the sensory characteristics of heat-treated camel milk. Comparing three treatment conditions (63 °C for 30 min, 72 °C for 15 s, and 100.5 °C for 10 min), Lund et al. [44] observed lower taste score, texture, and overall acceptability in treated samples of camel milk compared to control (nontreated); however, there was surprisingly no significant difference between pasteurization protocols, although the milk at 63 °C/30 min had the highest mean sensory score. However, these authors did not check the hygienic level of their samples. Indeed, to compare the different protocols, it is important to cross the data regarding pasteurization conditions and sensory characteristics with the hygienic level of the raw milk.

Some studies regarding behavior during heat treatment observed that camel milk could give an important amount of dry deposit on a stainless-steel plate during the pasteurization process from 60 to 90 °C for 1 h or 2 h [45]. This study showed that such a deposit is probably not of protein origin, because free thiol groups are in lower quantity than for cow milk treatment. A similar design of experiment with camel whey protein showed that, after 63 °C, whey proteins started to be denatured; this was especially evident at 98 °C [46]. It is possible that camel milk is producing a higher quantity of “milk stones”, i.e., the deposit of milk residues accumulated in an insufficiently cleansed dairy equipment where bacteria can be multiplied, contributing to bad flavors in milk. To the best of our knowledge, there are no available data in the literature regarding this aspect.

### 2.4. Camel Milk Protein Behavior Following Heat Treatment

In-depth studies on camel milk started relatively recently, especially regarding its technological properties. Even if some data on camel milk composition date from 1905 [47], research regarding the heat treatment impact on camel milk components started being implemented at the end of the 1980s. Thus, in a first trial, camel milk was heated at 63, 80, or 90 °C for 30 min. The rate of heat denaturation of whey proteins was twofold lower than for cow whey proteins [48]. After this study, another trial focused on heat coagulation at 100–130 °C in a pH range of 6.3–7.1. Only the 100 °C variant showed relative stability at pH 7, comparable to cow milk [49].

Because milk stability during heat treatment is the most important point and because the size of casein micelles could impact milk preservation in a homogeneous solution, the micelle size of camel milk was measured before and after pasteurization. They are broader than those of cow and human milk. However, an important point must be kept in mind, i.e., the samples were analyzed 36 h after sampling [50]. Regarding the mineralization and citrate quantity in camel casein micelles, the proportions were significantly different than for cow casein micelles after pasteurization. It was observed that, at pH 5, significant changes occurred in the dromedary casein micelle structure from milk to coagulum [51].

Some studies [35,52] stated that the pasteurization of camel milk could change its chemical composition. However, in these experiments, the microbial quality of raw camel milk was not assessed, no standardized starter culture was used, and the microbial contamination risk was not taken in account. Yet, in trials where microbiological status was tested before processing, pasteurization at 63 °C for 30 min improved the bacteriological quality of camel milk without changes in the composition compared to raw camel milk [36,53].

The observation of camel whey protein using SDS-PAGE showed that some proteins sensitively decrease after heat treatment, albeit to a lesser extent than for cow proteins. The pattern of camel whey proteins and the global composition of proteins are not the same for camel and cow milk. Accordingly, the major proteins of whey in bovine are serum albumin (SA), α-lactalbumin (α-La), and β-lactoglobulin (β-Lg), whereas those for camel are SA, α-La, and three other fractions not reported in cow milk [48,54,55,56,57].

In a recent study [58], the effect of heat treatment according to different protocols (65 °C for 30 min, 72 °C for 30 s, 75 °C for 5 min, 85 °C for 5 min, or 90 °C for 5 min) on camel whey proteins was assessed comparatively to cow milk. A lower denaturation of α-lactalbumin was observed in camel milk; after treatment at 90 °C for 5 min, 67% α-lactalbumin remained intact vs. only 5% in cow milk. However, camel serum albumin (CSA) was not totally detected after treatment at 85 °C for 5 min, although this was a less rapid process than with cow serum albumin, which disappeared after 75 °C for 5 min [58]. Similar figures were observed in a previous paper [45].

Some authors tried quantifying the changes in the concentration of camel whey proteins during the pasteurization process. However, debatable results occurred especially because different methods were used, needing clarification. Indeed, the quantification of whey proteins in camel milk can be done using electrophoresis [59], radial immunodiffusion [56,60,61,62], high-performance liquid chromatography [56], fast protein liquid chromatography [60], and the more recent techniques of liquid chromatography/tandem mass spectrometry and liquid chromatography/electrospray ionization mass spectroscopy [57].

The heat resistance of camel milk can also be revealed by the heat coagulation time [59]. Compositional difference plays an important role in heat resistance, particularly the absence of β-lg, different ratios in the casein complex (higher quantity of αs1-, αs2-, and β-caseins and lower quantity of κ-casein compared to cow milk [63]), and the presence of a higher quantity of other whey proteins. Whey proteins include three protein fractions described as common fractions of immunoglobulins (IgG1, IgG2, and IgG3), α-lactalbumin, lactophorin (which is closely related to the bovine proteose peptone component 3 (PP3)), the innate immunity peptido-glycan recognition protein (PGRP), and the whey acidic protein (WAP) [64].

### 2.5. Sterilized Milk

Sterilization of camel milk using an ultra-high-temperature (UHT) treatment is yet to be achieved, despite private companies trying to establish an appropriate method. Some studies on the heat resistance of whey and casein proteins, fat globules, vitamins, or other compounds of camel milk are expected to contribute to a technical solution [42,48,49,52,55,65,66,67].

After UHT treatment, camel milk presents a separation into two phases. During heat treatment at 65°/30 min, 72°/30 s, 75°/5 min, 85°/5 min, 90 °C/5 min [48] or at 63, 80 and 90 °C for 30 min and 72 °C for 15 s [52], it was observed that the whey proteins were overly sensitive and started denaturing. To stabilize camel milk proteins after UHT treatment, different protocols including the addition of chemicals (sodium hydroxide, calcium chloride, ĸ-casein from cow, sodium dihydrogen phosphate anhydrous, disodium hydrogen orthophosphate, or ethylenediaminetetraacetic salt) were tested but with disappointing results [68]. Further in-depth studies need to be implemented before being able to produce UHT camel milk.

However, it is possible to obtain sterilized milk after reconstitution of liquid milk using camel milk powder. This camel dairy product is available on the market in the Middle East. Moreover, ultrafiltration to yield concentrated skim camel milk was tested experimentally [69]. By using an α-alumina ultrafiltration membrane (pore size 1.4 µm) at an operating temperature of 50 °C, it was possible to obtain a product with an extended shelf-life up to 60 days and to decrease the germ count by 99%. The low filtration temperature did not damage the heat-sensitive proteins [70].

### 2.6. Antimicrobial Activity and Pasteurization of Camel Milk

Many people believe that camel milk has sufficient natural antimicrobial activity to preserve it from natural adulteration at ambient temperature for a long time. As such, they often consider that camel milk collection does not require the same requirements in terms of hygiene practices. Therefore, in many cases, potential microbial contamination is not checked before implementing the trials. Nevertheless, it is important to recall the results obtained by Sela et al. [71], whereby the thermal death time of *Escherichia coli* in camel milk was the same as in cow milk. The presence of *E. coli* in camel milk is common [72] and flouting hygienic rules will provoke digestive disorders in consumers. At the same time, raw and pasteurized camel milk has the capacity to inhibit *Cronobacter sakazakii*. As this bacterium can grow in powder milk, some authors think that it could be possible to use it in the production of infant formula [73], but such a possibility requires further investigation.

Obviously, pasteurization prolongs the shelf-life of milk and several data were obtained using different protocols. In the study of Lund et al. [44], quality was maintained after storage at ambient temperature for 24 h for raw milk vs. 76 h for milk treated at 100 °C/10 min, 64 h at 63 °C/30 min, and at 72 °C/15 s. The bacterial level appeared higher (2.77 log colony-forming units (CFU)/mL) with a protocol of 72 °C/15 s than 75 °C/10 min (2.65 log CFU/mL) and 65 °C/30 min (2.57 log CFU/mL), with the lowest content (2.45 log CFU/mL) being observed with a protocol of 80 °C/5 min [74].

In their study assessing the effect of two protocols (63 °C/30 min and 72 °C/15 s), on total bacterium, coliform, and mold counts, Mohamed and El-Zubeir [75] found that the first protocol was non-significantly more efficient in terms of total bacterial and coliform count, but less in terms of yeast and mold count (Figure 1). However, the protocols were applied on highly contaminated milk. Accordingly, it is worth reiterating that pasteurization does not guarantee sterilization and requires respecting the hygienic rules at the milking, storage, and transport stages of the raw material to correct process the milk.

Camel milk pasteurization is achieved at an industrial scale, and pasteurized camel milk can be directly provided to consumers with a shelf-life of around a week. Some dairy plants have extensive experience to producing pasteurized camel milk, for example, Tiviski (Mauritania), Camelicious (UAE), Al-Watania (KSA), and Tedjane (Algeria). However, further research is necessary, especially to study the rheological properties of raw and pasteurized camel milk, which are poorly documented.

## 3. Fermented Milk

The fermentation process is commonly used for the preservation of food. The process of milk fermentation, including camel milk, is a traditional ancestral method all over the world. It consists of the transformation of lactose into lactic acid by the natural microflora in milk dominated by lactic bacteria and, in some cases, by yeasts. To understand this process, numerous investigations on the microflora of raw camel milk were performed. As many fermented products from camel milk have been produced by spontaneous fermentation for centuries, we start our analysis of the literature by considering the global microflora (pathogenic or otherwise) of raw camel milk.

### 3.1. Microflora of Raw Camel Milk

#### 3.1.1. Nonpathogenic Microflora in Raw Milk

The normal microflora in camel milk has been widely investigated, and the results testify to high diversity across countries. In Tunisia, [76] found in raw camel milk 2.7 × 10^2^ CFU/mL of total mesophilic aerobic bacteria, 1.7 CFU/mL × 10^2^ of yeasts/molds, and 1.3 × 10^3^ CFU/mL of lactic acid bacteria (LAB). In total, 60 LAB strains were isolated. In camel milk samples from Morocco [77,78], 120 bacterial strains were isolated and identified as LAB by morphological and biochemical characterizations, as well as 16S ribosomal RNA (rRNA) gene amplicon sequencing. Among the described strains, some having probiotic properties were detected. Some of those strains were isolated in raw camel milk [79]. They belong to the genus *Lactobacillus* (*L. acidophilus, L. rhamnosus, L. gasseri,* and *L. delbrueckii*) [80]. Other strains belonging to the same genus (*L. plantarum, L. pentosus*, and *Lactococcus lactis*) were reported by Yateem et al. [81]. The diversity of microflora in camel milk also includes yeast as observed in Algeria where 12 species, dominated by *Trichosporon asahii, Pichia fermentans*, and *Rhodotorula mucilaginosa*, were identified [82].

Some strains isolated from raw camel milk such as *Leuconostoc mesenteroides* have shown specific antimicrobial activity against *Listeria* [83]. A similar finding was observed with *Enterococcus faecium* [84]. Abo-Amer [85] described the ability of *Lactobacillus acidophilus* strains isolated from raw camel milk to produce substances with antimicrobial activities. In a study achieved in Iran, among 64 LAB strains, 11 (belonging to genus *Enterococcus, Lactobacillus*, or *Pediococcus*) presented significant antibacterial activity against *Staphylococcus aureus* subsp. *aureus* or *Bacillus cereus* [86]. In Kenya, the predominant strains belonged to species *E. faecalis, S. agalactiae, Weisselia confuse*, *Rhodotorula mucilanginosa, Cryptococcus albidus,* and *Candida lusitaniae* [87,88]. In Kazakhstan, natural microflora including lactic bacteria (*E. durans, E. faecalis, E. faecium, Lb. casei, Lb. curvatus, Lb. kefiri, Lb. paracasei, Lb. sakei, Lc. lactis,* and *Lc. mesenteroides*) and yeasts/molds (*Kazakhstania unispora, Saccharomyces cerervisiae,* and *Kluyvermyces marxianus*) were identified in dromedary and Bactrian milks [89,90]. In Bactrian milk collected in China [91], 72 strains of lactic bacteria were isolated including *Lb. paracasei, E. italicus, E. durans, Lc. Lactis, W. confuse*, and *E. faecium.*

Such diversity has high technological interest, with the microflora playing important role both in terms of antimicrobial activity as emphasized above and in terms of acidification, which is essential for fermented products and cheese processing. However, due to the antimicrobial properties of camel milk proteins being greater than those in cow milk [92] and, in some cases, due to the low hygienic status of camel milk samples, the acidification process appears to be slower than for cow milk [93]. The starters used in camel milk processing for fermentation or cheesemaking (mesophilic, thermophilic, or their mixture) led to an acidification rate at 37 °C between 33% and 79% lower than for cow milk [88].

#### 3.1.2. Pathogenic Microflora of Raw Camel Milk

Substantial data have been dedicated to a preliminary bacteriological description of raw camel milk using different methods from all camel countries. Sela et al. [71] previously reported that camel milk could contain *E. coli* if elementary rules of hygiene were not applied. Camel milk could also contain other pathogenic strains such as *Streptococcus* or *Staphylococcus* species [94]. According to the latest study in Algeria, 58% of commercialized samples of raw camel milk were of satisfactory quality, 8.33% were acceptable, and 33.3% were unacceptable. However, in their samples, they did not find any *Salmonella* sp. and *Shigella* sp. Another study achieved in Sudan reported the presence of *E. coli*, *Klebsiella* spp., *Pseudomonas* spp., *Proteus* spp., *Enterococcus* spp., *Micrococcus* spp., *Streptococcus* spp., and *Staphylococcus* spp. [95]. The presence of coliforms can be observed despite the concurrent development of lactic acid bacteria. In Tunisia for example, a study on raw camel milk focused on the numeration of mesophilic count, total LAB, and coliforms, reporting values of 7 × 10^3^, 1.37 × 10^2^, and 1.8 × 10^1^ CFU/mL, respectively [96]. Therefore, it is important to check all raw camel milk samples for microbiological quality before processing. Indeed, when significant changes in camel milk composition are observed during the process, these could be linked to the microbiological status of the raw samples. The lack of systematic checks of microbiological quality in initial raw milk could impact all processing steps.

### 3.2. Diversity of Fermented Camel Milk

The microflora biodiversity leads to a rich diversity of fermented beverages prepared from camel milk. Moreover, fermentation is one of the oldest methods of consuming camel milk. Camel milk producers living in different regions of the world have their own varieties of fermented products with specific taste, texture, and flavor. Each camel country has described their traditional fermented milk in terms of microbiological, physicochemical, and chemical properties, as well as volatile organic compound profiles in some cases. The most known fermented camel milk products described in the literature are *shubat* in Kazakhstan [97] and China [98], *khoormog* in Mongolia [99], *garris* in Sudan [100,101], *suusac* in Kenya [102], *laben* (lben) in Arabic countries [103], and *ititu* and *dhanaan* in Ethiopia [104,105] (Table 2). Other traditional fermented beverages based on a mixture of camel milk and water are available in Mauritania known as *zrig* [106], in Morocco known as *Lfrik* [107], and in Iran and Turkmenistan known as *chal* [108].

However, in most cases, the fermentation process occurs spontaneously upon using previously fermented milk to inoculate raw milk [112]. The microflora in the fermented product is consequently more diversified than that in the raw milk samples [113]. For example, traditional *suusac* (Kenyan fermented camel milk) contains 45 LAB and three yeasts as identified by API50CHL and API20AUX. LABs were mainly represented by *Lactobacillus curvatus, Lb. plantarum, Lb. salivarius, Lactococcus raffinolactis*, and *Leuconostoc mesenteroides* subsp. *mesenteroides*, and yeasts were mainly represented by *Candida krusei, Geotrichum penicillatum*, and *Rhodotorula mucilaginosa* [111]. Other strains such as *E. faecalis, Lb. fermentum, Lc. lactis, Cryptococcus laurentii, Candida lusitaniae, Saccharomyces cerevisiae, Trichosporon mucoides*, and *T. cutaneum* were also identified more recently [88]. Traditional *garris* from Sudan contains the following LAB based on the API39CHL identification system: *Lactobacillus animalis, Lb. brevis, Lb. divergens, Lb. plantarum, Lb. rhamnosus, Lb. gasseri, Lb. paracasei, Lb. fermentum, Lactococcus raffinolactis*, and *Lc. alimentarium* [114].

Such a complex microflora ecosystem in fermented milk could lead to very variable final products, hardly compatible with obtaining a product with standard organoleptic quality. For example, *shubat*, which is prepared mainly in Kazakhstan, Uzbekistan, Russia, the Xinjiang region of China, and the western part of Mongolia, is made by spontaneous fermentation, often leading to the production of gas and foam and sometimes resulting in a particularly acidic product, provoking some reluctance in urban consumers [115]. Moreover, spontaneous fermentation can be affected by the presence of pathogenic strains of *E. coli* because the initial pH is not sufficient to suppress their growth [116]. To solve these problems, it is convenient to use starter cultures, i.e., a preparation containing a limited number of identified live microbial strains (single or mixed) inoculated in raw milk. Such management of controlled fermentation can lead to the expected sensorial properties of the final product by contributing to flavor and texture adapted to the urban consumers’ taste. This also contributes to a standardized and safer product on the market. Unfortunately, despite the high biodiversity of camel milk microflora, starter cultures used in the camel industry are mainly taken from bovine milk. In Kazakhstan, among the 104 isolates from *shubat* samples, sampled in different regions of the country, 79 were maintained in a pure culture (71 bacteria and 8 yeasts), while three of the strains were selected because of their biochemical properties (*Lactobacillus fermentum* К5, *Lactobacillus fermentum* К6, and *Lactobacillus plantarum* К7), before being tested for their growth kinetics and characteristics to measure their produced biomass [117,118]. This was important for developing an industrial culture in fermenters. Then, each culture was tested for technological suitability, acidity, bacterial colonization, and organoleptic properties [118]. The transfer to an industrial level was achieved thanks to the acquisition of a high-capacity bioreactor, wherein the media used were selected based on optimal bacterial growth and a convenient ratio of cost/growth potential. The final product (packaged starters in powder after lyophilization) was commercialized in two volumes (5 and 10 g per pack). However, industrial transfer remains difficult and requires supplementary studies regarding the technological properties of the numerous strains available in natural fermented milk [119]. In addition to the commercial interest for the development of fermented milk with standardized organoleptic properties, there is the public health benefit of such products thanks to their potential probiotic effect. For example, camel milk inoculated with LAB strain *Lactococcus lactis* KX881782 isolated from raw camel milk showed higher α-glucosidase inhibition (antidiabetic effect), antioxidant activity, angiotensin-converting enzyme inhibition (antihypertensive effect), and antiproliferative activity (anticancer effect) than cow milk [120].

## 4. Camel Cheese

Technical innovations regarding fermented camel milk have been applied to beverages known from prehistoric times to extend their shelf-life [121]. Furthermore, the making of camel cheese itself was an innovation. Indeed, the difference in casein proportions (notably the lower content of κ-casein) between cow and camel milk should explain the clotting difficulties observed in this process: 3–4% κ-casein vs. 13–15% in cow milk [122]. Moreover, bovine chymosin used in the dairy industry does not allow the optimal clotting of casein micelles from camel milk, leading to a weak curd. Thus, obtaining a firm coagulum was the first challenge faced by camel scientists and dairy factories processing camel milk [123].

### 4.1. The Challenge of Coagulation

The first trials were achieved in the 1980s using bovine rennet enriched in chloride and calcium phosphate with *Rhizomucor miehei* as coagulant (commercialized under the name of Camifloc^®^); however, the coagulum remained fragile and brittle [124]. Different vegetal coagulants were also tested as extracts from *Zingiber officinale* [125] or from *Moringa oleifera* [126], as well as abomasum extracts from young or adult camel [127]. The solution to getting a firm curd followed the sequencing of camel chymosin achieved by Kappeler et al. [122], which allowed introducing the coding gene for camel chymosin into a mold (*Aspergillus niger*). Later, this recombinant enzyme was produced by Chr. Hansen© at an industrial scale and marketed from 2008 under the trade name Chymax-M1000^®^. However, despite a good curd being a necessary condition, it did not guarantee a “good” cheese adapted to the consumers’ preference in terms of taste, especially because camel cheese was often generated by scientists in a laboratory rather than cheese technicians in dairy plants, except in Mauritania [128].

With great variety being possible in cheese, many trials are necessary to propose a large panel of products to the consumers. Different cheeses based on the technology for making gruyere [129], mozzarella [130], or feta and halloumi [131] were tested, but the texture, taste, and flavor of the final product did not correspond to the bovine equivalent. Indeed, the behavior of the camel’s “proteinic–lipidic matrix” during cheese processing differs from cattle milk. Such discrepancies among milk from different dairy species require more fundamental investigations of rheological properties to understand the changes during the different steps of acidification, coagulation, draining, brining, and refining, as well as the effect of various starters and thermal treatments [123].

### 4.2. Rheological and Microstructural Studies

As camel cheesemaking is a recent achievement all over the world, basic studies on the rheology or microstructure of the final products remain scarce. Investigations on curd tension and syneresis were achieved on soft camel cheese [132]. Gel firmness and gelation properties were tested according to different levels of temperature and pH [133]. The parameters of cheese viscosity (storage and loss moduli, loss tangent) after camel milk coagulation by camel chymosin were also reported [134]. To the best of our knowledge, while several studies on texture measurements (hardness, adhesiveness) have been published (for example, [135]), the microstructure properties of camel cheese require more investigations. In Iran, the effects of different mixtures of coagulants (*Rhizomucor miehei* protease and camel chymosin) on the microstructure and rheological properties of white cheese were compared, reporting a more compact protein network and firmer structure in the cheeses made with camel chymosin, but they used cow milk as a model [136]. In other studies, a comparison of the viscoelastic structure and the global gelation properties at different temperatures between cow and camel milk showed the higher difficulties of getting cheese with pasteurized camel milk [137].

However, only studies achieved at a laboratory scale are available. True investigations at the industrial scale are lacking. Moreover, in the published literature regarding the physicochemical and rheological characteristics of camel cheese, the microflora (pathogenic or otherwise) was not systematically considered.

### 4.3. Comparative “Behavior” with Cow Cheese

A comparison of the changes observed in the “proteinic–lipidic matrix” of camel and cow milk during cheese processing is a common feature in scientific literature to understand the specificity of their respective “behavior”. However, this comparison is of low interest when the gross composition of each specific milk differs significantly. To avoid this, Konuspayeva et al. [131] adjusted cow milk to obtain the same fat and total protein concentrations. Although they got similar cheese raw yields (7.4 ± 0.15 vs. 7.3 ± 0.55 kg/100 kg for camel and cow milk, respectively) and calcium recovery, camel cheese presented a higher recovery in terms of total nitrogen, while cow cheese retained more fat. Significant differences were also observed in lactoserum composition: camel lactoserum contained more fat and total nitrogen (9.0 ± 1.73 g/kg and 9.21 ± 0.23 g/kg, respectively) compared to cow (7.7 ± 1.61 g/kg and 7.30 ± 0.02 g/kg, respectively) despite similar dry matter (68.9% ± 3.2% and 68.1% ± 1.15% in camel and cow whey, respectively).

Two main technological difficulties are faced in camel cheese processing: (i) the continuous removal of serum from curd, and (ii) the slow acidification of the curd. Indeed, contrary to cow cheese where ripening is started after curdling with the rapid appearance of crust formation, camel cheese is characterized by a weak crusting and a continuous loss of moisture, due to serum release, leading often to a very dry curd which hinders correct ripening. Such behavior can be favorable for some types of cheese (such as feta), but not for hard cheeses [123]. This slow acidification was first reported by Farah and Bachmann [137], who found that 10 h was necessary to decrease the pH from 6.6 to 5. Such a delay can be reduced at 36 °C compared to 20 °C [130]. Thermophilic starters such as *Lactobacillus helveticus*, *L. lactis*, or *Streptococcus thermophilus*, known for their high acidifying power, can also be used to improve the acidification process [131]. However, for the management of fermentation to get specific fermented products, camel cheese manufacturing does not use starters made with lactic bacteria isolated from camel milk. The challenge for scientists and cheesemakers could be the identification of LAB strains specific to camel milk, allowing the potential to provide the typical aroma of cheese and to enrich the variety of camel cheese proposed to consumers.

Investigations on the interactions between minerals and camel cheese processing are also lacking. It has been established that there was no effect of the addition of phosphate or calcium chloride on improving gelation by camel chymosin [130] contrary to cow milk [138]. There is also a lack of information regarding camel cheese in terms of the analysis of volatile organic compounds responsible for aroma which supporting the typicity of the cheeses. Currently, such an analytical approach has only been developed for cow cheese [139].

### 4.4. The Challenge of Industrial Development

Up to now, it seems that scientists and cheesemakers aim to make cheese by using the same methodology as for cow cheese. However, when feta- or mozzarella-type cheeses are prepared using camel milk, the results are often disappointing because the taste, texture, and consistency differ with respect to the same cheese made using cow milk. In the Middle East, where consumers prefer products with a neutral taste, it is difficult to expect the development of cheeses with a strong character as found in western Europe. To achieve products adapted to the local consumers, different trials were proposed, for example, cheese spread [140] or white soft cheese [141,142]. However, few cheeses have given rise to sensory analyses assessing the acceptability of the product by the local consumers [131]. Yet, such research would be useful for the dairy industry to develop convenient cheeses. To the best of our knowledge, few industrial initiatives have seen the light of day [128].

The industrial development of camel cheesemaking is limited not only by the technological difficulties, but also by the hygienic quality of raw milk as mentioned above, notably because it is difficult to coagulate camel milk after pasteurization, forcing manufacturers to work with raw milk. The use of halloumi-type cheese technology [131] is an interesting alternative because the coagulum is pasteurized in lactoserum for 10 min at 80 °C after pressing. A second difficulty is linked to the cost of camel cheese due to the high price of the primary matter [143]. An alternative could be the valorization of lactoserum, which represents 88% to 90% of the initial milk volume.

## 5. Camel Milk Powder

As outlined in Section 1, camel milk is a recent development on the international dairy market, made possible by the development of powder milk production, which is the best way to preserve this highly perishable product for later consumption. Moreover, camel milk is often produced in remote places far away from the consumption basin, whereby the only solution to transport a high quantity of milk is by removing the water it contains (88% to 90% of the weight). Another advantage of this process is the conservation of the nutritive value of liquid milk. To make camel milk powder, two main modern technologies are used: spray-drying (hot-drying) and lyophilization (freeze-drying).

### 5.1. The Drying Technologies

The first reported trial aimed at making camel milk powder was recent, where freeze-drying technology was implemented to study the thermal characteristics of camel milk and its main components [7]. However, these tests were carried out in a laboratory (not at an industrial scale) with a freeze-dryer, allowing drying from −40 to 20 °C with a vacuum of 100 Pa. The resulting powder was stabilized at 11.3% humidity. A second, more recent study [144] pursued similar objectives, namely, to assess the effect of freeze-drying on the nutritional properties of camel milk, i.e., how the procedure affects the fine composition of camel milk in comparison to fresh milk. Analyses indicated a relative stability of most components (including minerals and vitamins) and concluded that the nutritional properties of camel milk powder were maintained. However, this was also achieved using laboratory equipment with limited capabilities. Moreover, nothing was said about the solubility of the powder obtained.

The spray-drying process was also the subject of a few scientific publications. In a study comparing the physical properties of powdered camel and cow milk obtained by spraying, [145] used a two-step process; milk was first concentrated to 20–30% dry matter using a rotating evaporator at 80 °C and then passed through a sprayer. The equipment used (FT80/81 Tall Form Spray Dryer) allows treating small amounts with the same effects of an industrial sprayer. In their protocol, the drying conditions were as follows: air intake temperature of 200–220 °C, air outlet temperature between 98 and 105 °C, pump speed at 3–5 arbitrary units, and air outlet humidity between 1.2% and 5.8%. In their conclusion, the authors indicated that this process allows obtaining powder with less than 1.8% water, thus allowing a long shelf-life. The drying temperature should be well controlled, as too high temperature results in an increase in the insolubility index due to protein denaturation. Overall, the solubility of camel milk powder is lower than that of cow milk. Another criterion used by manufacturers to assess the quality of milk powder is fluidity. This is the ratio between the density of the untamped powder and its density when compacted. This fluidity appears to be lower for camel milk compared to cow milk but remains at a fairly good level [145]. In a recent study regarding the acid gelation of fresh and powdered camel milk acid [146], the dry-spraying process was achieved using a laboratory sprayer (Buchi B-290), allowing an air entry temperature of 190 °C and an output temperature of 90 °C with an input flow of 600 mL/h. In their publication, [147] also aimed to test the effect of spray-drying, i.e., how the temperature of the air intake (160, 140 and 120 °C), atomization pressure (800, 600, and 400 bars), and feeding flow (5.4 and 3 arbitrary units/s) affect the nutritional components of camel milk (vitamin C, fatty-acid profiles). The equipment used was the same mini-laboratory sprayer cited in previous publications. Powder yield increased with the highest air intake temperatures and the lowest feed flows. High temperatures and high spray pressures reduced vitamin C levels. Lastly, the atomization pressure increased the fatty-acid content.

### 5.2. Interests and Limitations of the Technologies Used for Spraying Camel Milk

The spray-drying method seems preferable to make camel milk powder for a better reconstitution of liquid milk, but the investment for the dairy industry is more important as it requires the procurement of a costly milk drying tower and sprayer. However, the powder obtained by freeze-drying (lyophilization) could be used by agro-food industries (pastry and chocolate factories).

Nevertheless, the main limit is that drying consumes a high level of energy. In the dairy industry, the spray-drying process has a higher energy demand per ton of end-product, despite the recent technical improvements and novel equipment decreasing the energy consumption per ton of finished product. Moreover, the high investment necessary for obtaining and using a drying tower to make milk powder requires a sufficient volume of raw matter, which is only possible in certain contexts such as large camel dairy farms or collecting centers with a large network of camel farms.

Lastly, it appears that trials on camel milk (in any case, those that are published) are limited in number and that industrial trials are poorly documented. It is also apparent that the spraying of camel milk requires an optimization of the input parameters (temperature, pressure, and flow) to maintain the nutritional properties of the product and the functional characteristics (solubility, hygroscopy, and fluidity) of the powder. As camel milk studies were all carried out using materials dealing with small quantities, the “translation” of these results to an industrial scale involving large volumes is not possible, although the practice of industrial hot spraying of camel milk is implemented in the United Arab Emirates, China, and Europe.

### 5.3. The Challenge for Camel Milk Powder Development

The main problem with camel milk during high-temperature heat treatment, as happens during spraying, is the denaturation of proteins (especially whey proteins), which explains the difficulty with obtaining UHT milk. To maintain powder milk in the best possible conditions, and to facilitate the solubility of the powder to replenish the liquid milk, the surface composition of the powder is essential [148]. This surface of the spray-dried emulsion is naturally composed of mostly fat (mostly triglycerides), in addition to some proteins. Thus, the denaturation of serum proteins at a high temperature increases the fatty surface content of the powder and makes it difficult to replenish liquid milk. For a better emulsion during this reconstruction, it is proposed to perform an “encapsulation” using sodium caseinates, thereby ensuring stability of the powder. Such encapsulation is improved by the presence of lactose. For example, surface fat decreases from 30% to less than 5% if lactose is present in a 1:1 ratio relative to sodium caseinate.

In Table 3, possible improvements are reported to obtain a powder of stable quality due to the specificities of camel milk.

## 6. Other Products

### 6.1. Yogurt

Ample literature is available on the possibility of making yoghurt with camel milk [149,150]. Several strains of conventional lactic bacteria have been tested such as *Lactobacillus bulgaricus* or *Streptococcus thermophilus* [151], as well as *L. acidophilus*, *L. casei*, and *Bifidobacteria* [152].

However, the manufacture of camel milk yoghurt poses a texture problem, with the product appearing sticky and ultimately unpleasant to the palate [153]. Indeed, the viscosity of the product does not change during the gelling process compared to the milk of other dairy species. This constraint is related to protein composition [154] and to antibacterial factors naturally present in camel milk [155]. Another reason could be linked to the foaming properties of camel milk. The foam in this milk is stable, but it leads to a weak structure of the gel, which becomes unstable [156].

To obtain a better texture, trials with the addition of gelatin, alginate, or calcium were attempted [157], whereas ferments producing exo-polysaccharides were also used [158]. The application of a high-pressure treatment could have a positive effect on the texture, but no trials have been conducted to date with camel milk [159].

Other authors have attempted to improve the manufacture of camel milk yoghurt by mixing it with milk from other species [160] or by introducing 0.75% biosynthesized xanthan, albeit with moderate results in terms of organoleptic properties [161]. In any case, the final product corresponds at best to a “drinking yoghurt” without having the taste qualities, even when natural or synthetic aromas are added [162]. These difficulties explain why there is limited industrial production of camel milk yoghurt at present. Some researchers proposed frozen yogurt as a product that is between yogurt and ice cream [163]. The optimal composition from a texture point of view would be allowed with several ingredients such as fat (5%), sugar (13%), gelatin (0.5%), and 14% banana [164]; however, such a proposal has never gone beyond a laboratory scale.

### 6.2. Butter and Sweet

The fat in camel milk contains less than 0.5% butyric acid [165] compared to almost 5% in cow milk. In addition, fat cells are smaller than in cow milk [166]. As a result, butter yield is low [167] with disappointing organoleptic properties [167,168]. To obtain fat cells at the time of butter production, it is necessary to implement vigorous hot shaking (22–23 °C), which allows recovering about 80% of the fat [168]. Ghee (clarified butter), a popular product in India, has also been attempted using camel milk [169]; however, in addition to very low yield compared to buffalo or cow milk, the final product was found to be more susceptible to rancidity. Transformation into butter, therefore, does not seem fundamentally interesting in the context of an industrial valuation of camel milk. In fact, apart from trials in Ethiopia where butter consumption, including rancid butter for certain recipes, is popular, the production of camel butter has little future.

Making ice creams with different flavors is an easy technology. Ice cream made from camel milk is commercialized in the United Arab Emirates, Morocco, and Kazakhstan. The same technology is used as for other milks. Ice cream is highly popular among consumers and, above all, provokes less reluctance than other products. However, very few studies on the texture and sensory properties have been conducted [170].

There is no reference for processing camel milk into sweets. However, traditional products are available. For example, in Kazakhstan, a caramel called *Balkailmak* is obtained after a long thermal treatment of about 10 h at boiling temperature. The introduction of milk powder to chocolate as proposed in the Emirates can also be mentioned. A study on the acceptance of camel milk in a panel including 470 Emirati students showed a higher score with chocolate-flavored milk [171].

### 6.3. Non-Alimentary Processing of Camel-Milk

The manufacture of soaps and other cosmetic creams with camel milk is now common practice in many countries (Morocco, Mauritania, Saudi Arabia, India, Holland, China, Australia, etc.), whether on a semi-industrial scale or on a handicraft scale. China sells cabinets containing various cosmetic products from lipsticks to moisturizers, shampoos, and various lotions. The interest in the use of camel milk for the cosmetic industry benefits from the hypoallergenic properties of its proteins [172].

## 7. Conclusions

The “modernized” processing of camel milk is a recent feature compared to the milk from other dairy species. However, the technologies used to transform milk into pasteurized or fermented products, cheese or yoghurt, powder, or various sweets face two main challenges: (i) the systematic application of already proven technologies for cow milk is not necessarily suitable for camel milk and requires adaptations based on more fundamental research on the behavior of milk components during processing; (ii) the transfer of laboratory results already relatively numerous to an industrial scale remains insufficient, especially for products such as cheese or yoghurt, and it requires additional technical and economic analyses. The worldwide interest in camel milk, which is largely due to its expected health effect for consumers, is prompting basic research and development to continue investigations in order to translate technological innovations into products available on a large scale.

Despite these technological constraints, the global camel milk market is strongly changing. These changes are visible through two main structural innovations: (i) the emergence of intensive production systems under an entrepreneurial approach appearing more or less disconnected from pastoral dynamics; (ii) the development of periurban camel production systems, often initiated by pastoral breeders, contributing significantly to the urban supply in camel milk and keeping significant relationships with the pastoral economy. Such trends emphasize that the technical innovations in camel milk processing are coupled with socioeconomic dimensions. However, such changes raise the question of the sustainability of these new methods of producing milk and their environmental impact for the planet. Obviously, they contribute to a revaluation of the place of camel in national livestock economies, and it requires ensuring (i) the establishment of an international standard recognized by all the stakeholders, and (ii) the sustainability of “modernized” production and processing systems by avoiding the shortcomings noted in other modern livestock sectors.

## Figures and Tables

**Figure 1 animals-11-01045-f001:**
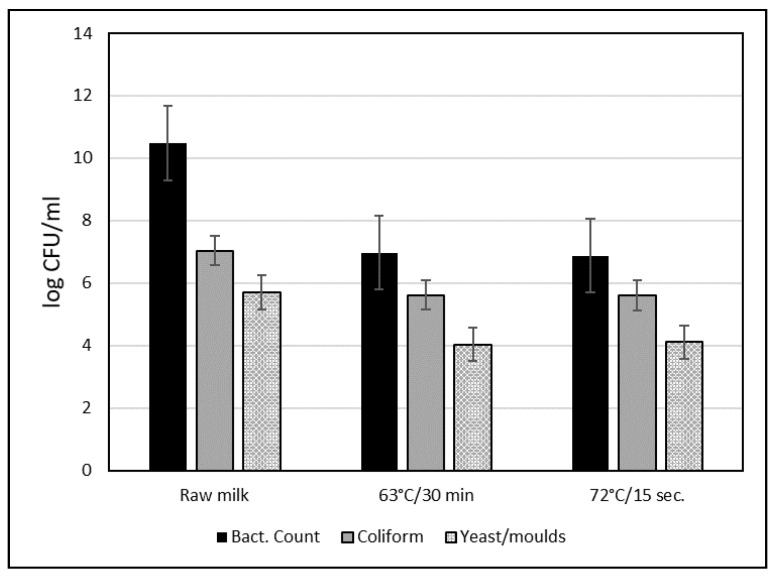
Changes in total bacterium, coliform, and yeast/mold count in camel milk before (raw) and after heat treatment (from [75]).

**Table 1 animals-11-01045-t001:** The main recent references (2019–2021) regarding camel milk processing. KSA, Kingdom of Saudi Arabia; UAE, United Arab Emirates; USA, United States of America.

Processing	Reference	Country	Topic
Pasteurization	Zhang et al., 2020 [7]	China	Fouling characterization
	Lajnaf et al., 2020 [8]	Tunisia	Foaming properties
	Li et al., 2020 [9]	China	Protein profile
	Yehia et al., 2020 [10]	KSA	Heat-resistant *Staphylococcus aureus*
	Bragason et al., 2020 [11]	Ethiopia	Antibacterial properties
Fermentation	Gammoh et al., 2020 [12]	Jordan	Modification of functional properties
	Sobti et al., 2021 [13]	UAE	Effect of pectin and alginate
	Mortazavi et al., 2021 [14]	Iran	Enrichment of fermented milk with pomegranate peel
	Zhadyra et al., 2021 [15]	China	Microflora diversity
	Sharma et al., 2021 [16]	KSA	Probiotic properties
	Ayyash et al., 2021 [17]	UAE	Exopolysaccharide impact
	Soleymanzadeh, 2019 [18]	Iran	Novel β-casein in fermented milk
	Edalati et al., 2019 [19]	Iran	Antagonistic effects of probiotic bacteria
Cheese	Mbye et al., 2020 [20]	UAE	Physicochemical properties, sensory quality, and coagulation behavior
	Belkheir et al., 2020 [21]	Algeria	Effect of starter on volatile and sensory profiles
	El-Hatmi et al., 2020 [22]	Tunisia	Cheese fortification by *Allium roseum*
Spray-dried and freeze-dried	Perusko et al., 2021 [23]	Serbia	Maillard reaction products
	Ho et al., 2019 [24]	Australia	Physicochemical properties with accelerated storage
	Deshval et al., 2020 [25]	India	Functional and morphological properties
	Zouari et al., 2020 [26]	Tunisia	Physical and biochemical properties
	Zouari et al., 2020 [27]	Tunisia	Microstructure and chemical composition
Yoghurt	Kamal-Eldin et al., 2020 [28]	UAE	Properties of mixed camel/cow yoghurt
	Buchilina and Aryana, 2021 [29]	USA	Yoghurt with monk fruit
	Atwaa et al., 2020 [30]	Egypt	Production of stirred camel yogurt
	Chen et al., 2019 [31]	China	Trisodium citrate and transglutaminase treatment of acid gel
Cream	Kashaninejad and Razavi, 2020 [32]	Iran	Properties and color

**Table 2 animals-11-01045-t002:** Some main characteristics of fermented camel milk from different countries.

Product	References	TVM	LAB	YM	Coli.	pH
		In log_10_ CFU/mL				
*Suusac*	Lore et al. [109]	9.03	6.77	2.05	1.00	6.0–4.25
	Jans et al. [110]	ND	7.2–8.5	ND	ND	4.9
*Garris*	Hassan et al. [35]	ND	ND	ND	ND	6.2–3.8
	Adelgadir et al. [111]	7.11–8.36	7.34–8.66	6.05–7.79	ND	3.79–4.32
*Shubat*	Rahman et al. [98]	ND	6.8–7.6	4.3–4.7	ND	ND

TVM: total viable microorganisms; LAB: lactic bacteria; YM: yeasts and molds; Coli: coliforms; CFU: colony-forming units; ND: not determined.

**Table 3 animals-11-01045-t003:** Possible improvements to obtain camel milk powder with high quality.

Process	Activity	Particularities	Comments
Raw camel milk (raw material)	Limit the number of suppliers or implement collecting centers with quality control	Low bacterial load (ideally <100 CFU/mL coliforms) Titratable acidity <16° Dornic	Without respect for hygiene, the milk may clot during powder processing and cause harmful fouling of the equipment
Concentration to remove some of the water (on average, there is 88% water in camel milk)	Remove at least 30% of the water using the principle of a “pressure cooker”, which is less expensive in terms of energy, saving atomization time and improving the quality of the final product	If hygienic standards are insufficient, risk of clotting	Clotted products are difficult to reuse, not only for the reconstruction of liquid milk, but also for the transformation into other products (cheese, yoghurt, fermented milk)
Homogenization to better emulsify	Make smaller fat and casein micelles to obtain a good-quality emulsion	Optional for better powder quality, a stabilizer can be added to the emulsion (caseinates), and lactose can possibly be removed to obtain an optimal 1:1 ratio of caseinates/lactose	This noncompulsory phase would result in a powder of better physical quality (solubility, fluidity, and low hygroscopic capacity) and stability over a longer period (preservation time)
Spray-drying	Optimize temperature (inlet/outlet), atomization pressure, and input flow to minimize energy costs and preserve the nutritional properties of camel milk	If hygienic standards are insufficient, the risk of appearance of poorly dried agglomerates will impact the commercial quality of the product	Data are available from suppliers
Packaging of camel milk powder	To preserve the quality of the finished product, it is advisable to use automated bagging	Working in a sterile atmosphere under strictly controlled humidity	This is an important step because any anomaly can involve all the previous steps (critical point in a Hazard Analysis Critical Control Point (HACCP) approach) Access to this part of the chain should be limited

## Data Availability

Not applicable.

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
