# Peer review of "Recent Advances in Camel Milk Processing"

_animals, 2021, doi:10.3390/ani11041045_

Round 1
Reviewer 1 Report
Comments : This review is a very complete synthesis of recent advances in the processing of camel milk, gathering a very important work with 143 references which are very well exploited in the document. This document gives an interesting and updated overview of the potential of camel milk in terms of processing compared to cow's milk and represents an interesting review for the journal Animals. Only minor points are mentioned below.
Minor points :
P2 L 73: please confirm if the pasteurization treatment scale is of 72°C for 20 min
P2 L 74: remove “also”
P2 L 74-75: remove: “other results were published. Thus”. But later in 2011, Lorenzen, et al. [14] concluded …
P2 L 77: change the number of the reference: Tayefi-Nasrabadi et al. [15], not [14]
P3 L 129-130: Please specify the concerned “structure” (dromedary casein micelle ?) i.e. It was observed that at pH 5, significant changes occurred in “the dromedary casein micelle (?)” structure from milk to coagulum.
P4 L 167: it seems more appropriate to reverse Chapters 2.5 and 2.6. Talk about pasteurization before talking about sterilization.
P4 L183: Moreover, ultrafiltration for making camel concentrated skim milk was tested [42]. This sentence is incomplete. The ultrafiltration approach should be further developed in terms of an interesting technique for the sterilization of camel milk.
P4 L 195: add “it” after use: to use it in the production of infant formula
P4 L 218: In the sentence : “However, further research is necessary, especially to study mainly the rheological properties of camel milk”, Does the authors mean camel milk or pasteurized camel milk ?
P4 L 219-220: A remark : the sentence “Further and in-depth studies need to be implemented before to be able producing UHT camel milk” can remain in this chapter if the next one deals with the UHT treatment.
P4 L 227: In the sentence “As many of fermented products from camel milk are made by spontaneous fermentation … ”, change by “As many fermented products from camel milk are produced by spontaneous fermentation …”
P4 L256: Change “identified in dromedary and Bactrian milk » by “were identified in dromedary and Bactrian milks”
P6 L 261: Added “fermented products” in the sentence “which is essential for fermented products and cheese processing
P6 L 273: Added “of”: “samples of camel milk were of satisfactory quality”
P6 L 275: Added “the”: “reported the presence of E. coli”
P6 L 275: added the superscript number : values of 7x103; 1.37x102 and 1.8x101 – (7x103 …)
P8 L 333: Added “of”: “selected because of their biochemical properties”
P8 L 381: In the sentence: “coagulation, acidification, draining”, change in “acidification, coagulation, draining”
P9 L 403: Complete the title as: “Comparative “behavior” with cow cheese
P11 L534-536: Precise which surface: “This surface is naturally composed mainly of fat”. Surface of a grain of powder?
Explain also what the fatty surface part of the powder is, and how denatured serum proteins could increase the fatty surface part of … “:
Author Response
We thank the reviewing suggestions that we added in the reviewed version
Reviewer 2 Report
This is a review paper that covers an excessively broad topic. Consequently, it is not deep enough on some issues. There are also too many non-referenced affirmations, which are certainly interesting, but not what one expects in a bibliographic review. It is more of a technical text than a scientific one in parts. I suggest that the authors focus the review on a more concrete topic that allows them to successfully cover all the current scientific baggage.
Minor comments:
- Line 156. Define abbreviations the first time they are used. This is applicable to the entire document.
- Line 354. Scientific names in italics. Review throughout the document.
- Table 2. Possible improvements must be supported by references, and properly explained.
Author Response
We thank the rewiewing suggestions and we added the corrections. Regarding the remark on the"more concrete topic", we would like to know what it's mean. Because if we focus on more concrete topic, the paper will be still more technical than scientific, reversely to what is expecting by the reviewer. In contrary, in our paper, we try to recall not only the technical aspects but all the challenges to be met by the science for improving the camel milk processing.
Reviewer 3 Report
The manuscript entitled "Recent advances in camel milk processing" presented the review on the benefits and nutritional importance of products made from camel milk. This paper is interesting to diversify dairy products and to use alternative milk sources. Generally speaking, the review is interesting and deserve to be published. Moreover, the purpose of manuscript is clear and paper is logically structured. However, a thorough revision of English for grammar, syntax, vocabulary is still necessary.Author Response
We thank the reviewer for his suggestions. We tried to improve English with external support
Reviewer 4 Report
This is an excellent review and very well written in excellent English. It is showing many shortcomings in the literature of camel milk and makes an excellent new contribution for any new business people who want to get into the camel milk market. Actually a more detailed discussion of the biochemistry of camel milk would be a welcome improvement to the paper, which would better explain the weakness of camel milk in the bovine traditional processing industry.
Author Response
According to the suggestion of the reviewer, we added a short paragraph on camel milk composition.